# PLIN: A Network for Pseudo-LiDAR Point Cloud Interpolation

**DOI:** 10.3390/s20061573

**Published:** 2020-03-12

**Authors:** Haojie Liu, Kang Liao, Chunyu Lin, Yao Zhao, Meiqin Liu

**Affiliations:** Beijing Key Laboratory of Advanced Information Science and Network, Institute of Information Science, Beijing Jiaotong University, Beijing 100044, China; hj_liu@bjtu.edu.cn (H.L.); kang_liao@bjtu.edu.cn (K.L.); yzhao@bjtu.edu.cn (Y.Z.); mqliu@bjtu.edu.cn (M.L.)

**Keywords:** 3D point cloud, pseudo-LiDAR interpolation, convolutional neural networks, depth completion, video interpolation

## Abstract

LiDAR sensors can provide dependable 3D spatial information at a low frequency (around 10 Hz) and have been widely applied in the field of autonomous driving and unmanned aerial vehicle (UAV). However, the camera with a higher frequency (around 20 Hz) has to be decreased so as to match with LiDAR in a multi-sensor system. In this paper, we propose a novel Pseudo-LiDAR interpolation network (PLIN) to increase the frequency of LiDAR sensor data. PLIN can generate temporally and spatially high-quality point cloud sequences to match the high frequency of cameras. To achieve this goal, we design a coarse interpolation stage guided by consecutive sparse depth maps and motion relationship. We also propose a refined interpolation stage guided by the realistic scene. Using this coarse-to-fine cascade structure, our method can progressively perceive multi-modal information and generate accurate intermediate point clouds. To the best of our knowledge, this is the first deep framework for Pseudo-LiDAR point cloud interpolation, which shows appealing applications in navigation systems equipped with LiDAR and cameras. Experimental results demonstrate that PLIN achieves promising performance on the KITTI dataset, significantly outperforming the traditional interpolation method and the state-of-the-art video interpolation technique.

## 1. Introduction

In many computer vision tasks, dense and precise depth information of an outdoor environment is extremely important for various applications of autonomous driving and robotics. Recently, in 3D object detection [1,2,3], 3D semantic segmentation [4,5,6], and depth completion tasks [7,8,9,10], point clouds obtained by LiDAR have gained more and more attention due to their accurate spatial information. However, LiDAR sensors suffer from a low frequency for an autonomous driving system. Therefore, there is a mismatching time stamp between the LiDAR and other sensors such as cameras. In order to match the scenes collected by these two sensors and achieve system synchronization, the frequency of the camera has to be decreased to that of the LiDAR sensor, which significantly wastes resources and results in inferior performance for high speed applications. Therefore, it is quite appealing for autonomous driving systems to increase the frequency of its LiDAR sensor, and further pursuit the high-quality synchronized perception of a multi-sensor system.

To address the above problem, one possible solution is to interpolate an intermediate point cloud using two consecutive point clouds. However, directly working on the 3D space and generating a new point cloud is challenging. Due to the disorder and sparseness of point clouds, it requires huge computing resources in 3D space and it is difficult for neural networks to learn the generation of point clouds in large-scale scenes. Therefore, previous researches [11,12,13,14] prefer to achieve different tasks on 2D depth maps or other projected views. Moreover, the target point cloud can be constructed in the form of Pseudo-LiDAR [15] using known camera intrinsics. For Pseudo-LiDAR interpolation, an intermediate depth map is first generated and then back-projected into the 3D space. This method is superior to direct point cloud interpolation methods in term of feasibility and efficiency.

Interpolation techniques have been widely used in lots of computer vision and robotics tasks, which can be classified into two categories, i.e., temporal interpolation [16,17,18] and spatial interpolation [7,19,20]. The typical representative of temporal interpolation is video interpolation. In video processing, video interpolation aims to temporally generate an intermediate frame using two consecutive frames. This technique has attracted more attentions due to the increasing demand for high-quality slow-motion videos. However, this temporal interpolation technique has not yet been investigated in the field of point cloud interpolation. In contrast to video interpolation, depth completion spatially fills missing depth values in a sparse depth map to generate a dense depth map. This technique becomes an essential enhancement process for LiDARs as they usually only provide sparse measurements.

Motivated by aforementioned studies, we propose a Pseudo-LiDAR point cloud interpolation network (PLIN) to generate both temporally and spatially high-quality point cloud sequences. The overall pipeline of the proposed method is illustrated in Figure 1. The first two rows of Figure 1 show inputs of our network. The dense depth map of the third row shows the output of the network. The last two rows of Figure 1 are the final output form after the transformation module. Specifically, PLIN consists of a motion guidance module, a scene guidance module, and a transformation module. In the motion guidance module, we obtain bidirectional optical flow maps from color images, then warp two sparse depth maps into an approximate intermediate depth map. The original and warped depth maps, as well as the estimated optical flow maps, are fed into a coarse interpolation network to generate a coarse intermediate depth map. Subsequently, to produce a more accurate and dense depth map, we design a refined interpolation network with the guidance of realistic scenes. This scene guidance module generates a refined depth map using the intermediate color image and estimated depth map. Finally, in the transformation module, the refined depth map is used to construct the target Pseudo-LiDAR point cloud using camera parameters. Compared with video interpolation and depth completion tasks, our method simultaneously performs interpolation in both spatial and temporal domains. To the best of our knowledge, this is the first deep framework for Pseudo-LiDAR point cloud interpolation. Experimental results demonstrate that PLIN achieves promising performance on the KITTI [21] dataset.

In summary, we conclude the following three contributions of the proposed method:To mitigate the low frequency limitation of LiDAR sensors, we present a Pseudo-LiDAR interpolation network to generate both temporally and spatially high-quality point cloud sequences.We use the bidirectional optical flow as an explicit motion guidance for interpolation. In addition, a warping layer is applied to improve the accuracy of depth prediction by approximating an intermediate frame. Finally, the in-between color image is leveraged to provide rich texture information of the realistic scene for more accurate and dense spatial reconstruction.We evaluate the proposed model on the KITTI benchmark [21], which reasonably recovers the original intermediate 3D scene and outperforms other interpolation methods.

## 2. Related Research

In this section, we briefly review the techniques of the video interpolation and depth completion, which provide the inspirations of the temporal and spatial interpolations for our work, respectively.

### 2.1. Video Interpolation

In the application of the video processing, due to the increasing demand for high-quality slow-motion videos, video interpolation has been increasingly studied. The task of video interpolation tries to generate the intermediate frame using two consecutive frames, in terms of the accurate optical flow estimation. For example, Peleg et al. [16] formulated the interpolated motion estimation problem as classification rather than regression. This method achieves real-time temporal interpolation for high resolution videos. Bao et al. [17] perceived both the depth and flow information to address the strong occlusion problem during new frame synthesis. To interpolate more in-between frames, Jiang et al. [18] proposed a variable-length multi-frame interpolation method to generate a frame at any time step between two given frames.

### 2.2. Depth Completion

Depth completion tasks are often guided by color images and use relatively sparse depth maps to predict high-resolution dense depth maps. The modal of depth completion input data is mainly: Relatively dense depth input [22] collected by structured light sensor and sparse depth measurement [23] collected by LiDAR. Depth completion for relatively dense depth input often refers to depth enhancement [22] or depth inpainting [24], which is to interpolation the missing depth on a relatively dense depth map. When the input becomes 3D LiDAR where the projected depth measurements on the image space account for roughly 4% pixels [25], this problem will become more challenging. In earlier work [26], wavelet analysis was used to generate dense depth or diaparity. Recently, deep learning-based methods have achieved better performance in deep completion tasks. For example, Ma et al. [7] fed the concatenation of a sparse depth map and a color image into an encoder-decoder network to produce a dense depth map using self-supervised learning. In order to improve the performance of deep completion tasks, Zou et al. [8] designed a multi-task learning framework sharing a common encoder part and introduced boundary features as internal constraints in the decoder part. To obtain more accurate dense depth maps, Zhang et al. [19] employed a weight matrix to describe a surface normal and occlusion boundary. To perceive both surface normal and contextual information, Lee et al. [20] presented an end-to-end convolutional neural network for depth completion, which consists of a geometry network and a context network. Compared to these deep completion tasks that focus on spatial densification, our task is dedicated to spatially and temporally interpolating LiDAR point clouds. Temporal information, spatial information, and scene motion relationships are gradually utilized to generate temporally and spatially high-quality point cloud sequences.

## 3. Approach

In this section, we describe the detailed architecture of the proposed Pseudo-LiDAR interpolation network (PLIN). Given three consecutive color images captured by a camera and two depth maps obtained by LiDAR, we first interpolate an intermediate 2D dense depth map and then back-project the depth map into Pseudo-LiDAR using prior camera intrinsics. In addition, we explore the guidance of motion and scene to generate a realistic dense depth map, and adopt a warping layer to improve the accuracy of spatial reconstruction. Although the 2D dense depth map is the output of our refined interpolation network and is very important, our final result is point cloud form after the transformation module. Figure 2 shows the framework structure and overall pipeline of the proposed Pseudo-LiDAR interpolation network in detail. As illustrated in Figure 2, PLIN consists of a motion guidance module, a scene guidance module, and a transformation module. As a benefit of this coarse-to-fine cascade structure, our method can progressively perceive the multi-modal information and generate a temporally and spatially high-quality point cloud sequence. Moreover, we introduce the whole training loss function of PLIN in this section.

### 3.1. Intermediate Depth Map Interpolation

In this part, we introduce the method for intermediate depth map synthesis. We first present a baseline network to generate an interpolation map using only two consecutive sparse depth maps. Then, to construct more reasonable slow-motion results, we use the motion information included in a bidirectional optical flow to guide the interpolation process. Moreover, a warping operation is applied to input depth maps to produce an intermediate coarse depth map, which contains the explicit motion relationship. Finally, we use the in-between color image to refine the coarse depth map with the guidance of the scene, resulting in a more accurate and dense intermediate depth map.

#### 3.1.1. Baseline Network

As mentioned in Section 1, due to challenges of 3D point clouds, previous works [11,12,13] perform different vision tasks on 2D depth maps or other projected views. Inspired by this principle, we first interpolate an intermediate depth map using two consecutive depth maps.

Given two sparse depth maps $d_{t-1}$ and $d_{t+1}$, our goal is to synthesize a depth map $d_{t}$ for the intermediate frame. A straightforward way is to train a baseline network (i.e., an encoder-decoder structure) to predict the depth value of each pixel in the intermediate frame. Specifically, the encoder consists of a set of convolutions to increase the number of channels and reduce the feature resolution. The decoder has a symmetric structure. Moreover, to compensate the interaction of different information, the network structure contains multiple skip connections to combine low-level and high-level features at the same spatial resolution. Before feeding into the encoder, consecutive sparse depth maps are processed using a convolutional layer with eight 3 × 3 kernels and these extracted features are concatenated as the input to the network. All convolutions are followed by a batch normalization and a ReLU layer in the baseline network, with the exception of the last convolution layer, where a linear activation function is used. In the encoder part, we use ResNet-34 [27] as our backbone. In the decoder part, five fractionally-strided convolutional layers are designed to increase the resolution of image. After these five convolutions layers, a multi-channel feature map is obtained. The feature map is then passed through a 1 × 1 convolution kernel to generate a single-channel depth map. Thus, the intermediate dense depth map derived from the baseline network can be expressed as:(1)$$d_{t}=H^{b}\left(d_{t-1},d_{t+1}\right),$$
where the depth map $d_{t-1}$ is the previous frame depth map, $d_{t+1}$ is the latter frame depth map, and $H^{b}$ is an interpolation function learned by the baseline network. This formula shows that our baseline network can obtain dense depth map of intermediate frame by inputting adjacent sparse depth maps.

The baseline network adopts a violent approach to learn the relationship between the intermediate depth map and adjacent depth maps without any other guidances. However, due to the sparsity of the data, the complex motion relationship among $d_{t}$, $d_{t-1}$, and $d_{t+1}$ is difficult to estimate using only depth maps, and thus the obtained result usually shows an inferior appearance with blur artifacts. The results generated by the baseline network are shown in Section 4.2. For the interpolation task, neural networks should not only learn to generate the appearance of two input depth data distributions, but also accurately perceive the motion relationship among consecutive depth maps. In order to achieve more reasonable interpolation results, we introduce optical flow to guide the generation of dense depth maps.

#### 3.1.2. Motion Guidance Module

To consider the motion relationship between consecutive sparse depth maps, we design a motion guidance module to exploit optical flow to explicitly guide the generation of dense depth maps. In the video interpolation task, the optical flow is often used as an important input component, because it represents the direction and level of motion. Inspired by video interpolation, we introduce the optical flow into our Pseudo-LiDAR interpolation problem.

Instead of directly investigating the optical flow on sparse depth maps, we learn the motion relationship on dense color images due to their abundant and precise contextual information. Recently, deep neural networks have shown excellent performance in optical flow estimation [28,29,30]. Given two consecutive color images $C_{t-1}$ and $C_{t+1}$, video interpolation [16,17,18] aims to generate the intermediate color image $C_{t}$ using a bidirectional optical flow:(2)$$\begin{array}{c} F_{t-1\to t+1}=H^{f}\left(C_{t-1},C_{t+1}\right),\\ F_{t+1\to t-1}=H^{f}\left(C_{t+1},C_{t-1}\right), \end{array}$$
where $C_{t-1}$ is the color image of previous frame, $C_{t+1}$ is the next frame, and $H^{f}$ is an optical flow estimation function learned through neural networks. This formula shows that our optical flow estimation network can obtain bidirectional optical flow through inputting adjacent color images. Assuming that the motion of adjacent frames is smooth, the optical flow $F_{t\to t-1}$ for color images $C_{t}$ and $C_{t-1}$, and the optical flow $F_{t\to t+1}$ for color images $C_{t}$ and $C_{t+1}$ can be calculated as follows.
(3)$$\begin{array}{cc} F_{t\to t-1} & =0.5F_{t+1\to t-1},\\ F_{t\to t+1} & =0.5F_{t-1\to t+1} \end{array}$$
or
(4)$$\begin{array}{cc} F_{t\to t-1} & =-0.5F_{t-1\to t+1},\\ F_{t\to t+1} & =-0.5F_{t+1\to t-1} \end{array}$$

Equations (Equation 3) and (Equation 4) can be further combined into the following equation:(5)$$\begin{array}{cc} F_{t\to t-1} & =-0.25F_{t-1\to t+1}+0.25F_{t+1\to t-1}\\ F_{t\to t+1} & =0.25F_{t-1\to t+1}-0.25F_{t+1\to t-1} \end{array}$$

Different from the aforementioned video interpolation task, we devote to the generation of an intermediate point cloud. Note that the intermediate color image $C_{t}$ is available in this problem because the frequency of the camera is higher than that of LiDAR. Therefore, we can easily get $F_{t\to t-1}$ and $F_{t\to t+1}$ using the optical flow estimation network:(6)$$\begin{array}{c} F_{t\to t-1}=H^{f}\left(C_{t},C_{t-1}\right),\\ F_{t\to t+1}=H^{f}\left(C_{t},C_{t+1}\right). \end{array}$$

Considering that the LiteFlowNet [28] outperforms FlowNet2 [30] in terms of KITTI benchmarks [21] and the challenging Sintel final pass [31]. And the LiteFlowNet has the advantages of small model size and fast running speed. Therefore, we exploit the LiteFlowNet to estimate the optical flow of the consecutive color images. The generation of the intermediate depth map under the motion guidance can be expressed as:(7)$$d_{t}=H^{m}\left(d_{t-1},d_{t+1},F_{t\to t-1},F_{t\to t+1}\right),$$
where $d_{t-1}$ and $d_{t+1}$ refer to adjacent sparse depth maps, $F_{t\to t-1}$ and $F_{t\to t+1}$ refer to bidirectional optical flow, and $H^{m}$ is a depth map interpolation function learned through the motion guidance module. This formula represents that adjacent depth maps and bidirectional optical flow are used as inputs to the motion guidance module to obtain dense depth maps.

To make full use of the information provided by the optical flow, we leverage the bidirectional optical flow to directly produce an approximate intermediate depth map ${\hat{d}}_{t}$, in terms of the warping operation:(8)$${\hat{d}}_{t}=\gamma \cdot \;warp\left(d_{t-1},F_{t\to t-1}\right)+\left(1-\gamma \right)\cdot \;warp\left(d_{t+1},F_{t\to t+1}\right),$$
where γ refers to the weighting factor of two input depth maps and warp is a backward warping function that can be implemented using bilinear interpolation [18,32]. This warping layer transfers the depth map of adjacent frames to the position of intermediate frame using the estimated optical flow. Instead of roughly feeding the optical flow into neural networks, we further utilize the explicit motion relationship to build an approximate intermediate depth map, contributing to a more accurate 3D reconstruction.

Therefore, the input of the motion guidance module includes the two consecutive sparse depth maps, the estimated bidirectional optical flow, and the warped intermediate depth maps. The final interpolated intermediate depth map $d_{t}$ can be formulated by:(9)$$\begin{array}{c} d_{t}=H^{m}(d_{t-1},F_{t\to t-1},warp\left(d_{t-1},F_{t\to t-1}\right),\\ d_{t+1},F_{t\to t+1},warp\left(d_{t+1},F_{t\to t+1}\right)). \end{array}$$

#### 3.1.3. Scene Guidance Module

In contrary to the sparse point cloud, color images have richer and denser texture information, which significantly boosts the complete scene understanding. In order to obtain more precise and dense interpolation results, we design a scene guidance module to refine the coarse depth map $d_{t}^{c}$ derived by the motion guidance module. We first utilize two convolutional layers with the channels of 8 to extract features of the coarse depth map and color image, respectively. Subsequently, the convolved features are fused to form the input of the refined interpolation network. The refined interpolation network is a lightweight U-Net structure [33], the number of its layers is less than that of the coarse interpolation network. Specifically, the encoder contains five convolutional layers and the decoder contains four deconvolutional layers. The batch normalization and ReLU activation function are implemented to all convolutional and deconvolutional layers, expect for the last deconvolutional layer that uses the linear activation function. In addition, there are skip connections between feature maps with the same spatial resolution, to facilitate the complementation of local and global information. Thus, the intermediate dense depth map generated by the scene guidance module $H^{s}$ can be expressed as:(10)$$d_{t}=H^{s}\left(d_{t}^{c},C_{t}\right).$$
where $C_{t}$ refers to the intermediate color image, $d_{t}^{c}$ is the coarse depth map obtained by the coarse interpolation network, and $H^{s}$ is a depth map interpolation function learned through the scene guidance module. This formula shows that a coarse depth map is guided by a color image to generate a refined dense depth map $d_{t}$.

### 3.2. Transformation Module

Our final result is the form of a point cloud, so we need to perform dimensional transformation on the dense 2D depth map. Once intermediate depth map is generated, the point cloud can be constructed in the form of the Pseudo-LiDAR. According to the pinhole camera model principle, each spatial point (x,y,z) is corresponded to its pixel coordinates (u,v,d), where *d* refers to depth value. Through these camera parameters, the interpolated depth map $d_{t}$ can be converted into the coordinates of a point cloud. Here, we can derive the 3D position (x,y,z) of each pixel (u,v) in the camera coordinate system as
(11)$$\begin{array}{cc} z & =d_{t}\left(u,v\right), \end{array}$$
(12)$$\begin{array}{cc} x & =\frac{\left(u-c_{u}\right)\times z}{f_{u}}, \end{array}$$
(13)$$\begin{array}{cc} y & =\frac{\left(v-c_{v}\right)\times z}{f_{v}}, \end{array}$$
where $\left(c_{u},c_{v}\right)$ is the pixel position corresponding to the center of camera aperture, and $f_{v}$ and $f_{u}$ are the vertical and horizontal focal lengths, respectively.

By converting all pixels in the depth map into 3D coordinates, we can get a set of points ${\left(x_{i},y_{i},z_{i}\right)}_{i=1}^{n}$, where *n* is the number of points. The point cloud obtained from the intermediate depth map is named as Pseudo-LiDAR [15].

### 3.3. Loss Function

The whole loss function of PLIN is a linear combination of the coarse depth loss and the refined depth loss. The ground truth depth map of the intermediate frame can be used to supervise the network prediction. We adopt L2 Loss between the predicted dense depth map pred and the ground truth gt as follows
(14)$$L_{d}\left(pred,gt\right)={∥1_{\left\{gt>0\right\}}\cdot \left(pred-gt\right)∥}_{2}^{2}.$$

Our final loss function can be expressed as follows:(15)$$L=w_{1}\cdot L_{d}\left(d_{coarse},gt\right)+w_{2}\cdot L_{d}\left(d_{refined},gt\right),$$
where $d_{coarse}$ refers to the intermediate depth map of the coarse interpolation network, $d_{refined}$ refers to the intermediate dense depth map of the refined interpolation network, $w_{1}$ and $w_{2}$ are weights to balance two different loss functions. We pay more attention to the refined depth map, so we set $w_{2}$ to 1. In the experiments, we set $w_{1}$ to 0.01, 0.05, 0.1, 0.15, 0.2 respectively. We noticed that the performance is best when $w_{1}$ is 0.1. Therefore, in this work, $w_{1}$ and $w_{2}$ are empirically set to 0.1 and 1, respectively.

## 4. Experiments

In this section, we first describe the training dataset and strategy of the proposed PLIN network. We then perform several ablation experiments to verify the effectiveness of different modules in our network. To demonstrate its superiority, we also compare our method with a traditional method and an advanced video interpolation method. Figure 3 shows eight examples of interpolated depth maps. This figure is to depict our method has the ability to interpolate the dense depth map of the intermediate frame. In Figure 3 the color image is part of input, the relatively dense depth map in column 3 is our ground truth, and the last column is our network predicted dense depth map. As illustrated in Figure 3, the depth maps obtained by our method show clear boundaries in visual effects and display denser distributions than the ground truth dense depth maps.

### 4.1. Dataset and Strategy

#### 4.1.1. Dataset

The main application scenario of our model is on-board LiDARs for outdoor scenes. Our experiments were performed on the KITTI depth completion dataset [25] and the raw data dataset [21]. The KITTI dataset provides depth information and color images. The dataset contains 85,898 training data, 6852 validation data, and 1000 test data. Considering that the training set contains some frame sequences with tiny motion, we select 40,000 scenes with relatively large motion to train our network.

#### 4.1.2. Strategy

Since the upper part of the depth map of LiDAR projection does not provide any depth information, our network takes images with 1216 × 256 by bottom-cropping on original images. In addition, we perform data augmentation operations such as random flipping and color adjustment on training data. The whole network was trained in an end-to-end manner. We used the Adam optimizer with an initial learning rate of $10^{-5}$, and the learning rate was dropped by a factor of 0.1 after every five epochs. Our network was implemented in PyTorch [34] with a batch size of 1 and trained on a 1080Ti GPU for about 60 h.

### 4.2. Ablation Study

To evaluate the effectiveness of each module, we perform ablation study on the proposed network. Firstly, we conduct three experiments on the coarse interpolation network. Then, there are two experiments on the refined interpolation network as follows.

The baseline network only takes two consecutive sparse depth maps as the input (baseline).The forward and backward sparse depth maps and estimated optical flow maps are fed into the baseline network (baseline + flow).The baseline network receives the forward and backward depth maps, the bidirectional optical flow, and the depth maps derived by the warping layer (baseline + warp_flow).The refined network takes the intermediate color image and two depth maps as its inputs (baseline + rgb).The complete configurations including the coarse interpolation network with motion guidance using the warping operation, and the refined interpolation network with scene guidance (ours).

For the evaluation of interpolated depth maps, we choose four metrics: the root mean square error [mm] (RMSE), mean absolute error [mm] (MAE), root mean square error inverse depth [1/km] (iRMSE), and mean absolute error inverse depth [1/km] (iMAE). Similar to [7,19,20], we primarily focus on RMSE, which is the leading metric on the depth completion benchmark. Given ground truth depth $d_{gt}$ and complete depth $d_{pred}$, the metrics can be defined as the following formula:Root mean squared error (RMSE):
(16)$$RMSE=\sqrt{\frac{1}{n}\sum {\left(d_{pred}-d_{gt}\right)}^{2}}$$Mean absolute error (MAE):
(17)$$MAE=\frac{1}{n}\sum \left|d_{pred}-d_{gt}\right|$$Root mean squared error of the inverse depth [1/km](iRMSE):
(18)$$iRMSE=\sqrt{\frac{1}{n}\sum {\left(\frac{1}{d_{pred}}-\frac{1}{d_{gt}}\right)}^{2}}$$Mean absolute error of the inverse depth [1/km](iMAE):
(19)$$iMAE=\frac{1}{n}\sum \left|\frac{1}{d_{pred}}-\frac{1}{d_{gt}}\right|$$

The results of ablation study are listed in Table 1. The ablation study shows that the complete network (ours) achieves the best interpolation performance. For each module of PLIN, due to the provided optical flow between consecutive frames, the baseline network achieves more accurate results with the guidance of motion. Moreover, compared with the direct use of optical flow, the warping layer significantly improves the performance of interpolation, because of the more explicit intermediate representation. As a benefit of the rich texture information in color images, the baseline network with the guidance of scene outperforms the baseline network that only takes two consecutive depth maps as inputs. For the other minor evaluation metrics, the motion guidance module slightly increases their values, as the estimated optical flow obtained by LiteFlowNet [28] contains some noises. To intuitively compare these different performances, we visualize the interpolated results of a scene obtained by the above methods in Figure 4. Since other results have the same trend, in order to save space, we only show two groups of examples here. Figure 4 shows two examples of the ablation study. For each example, the first row of Figure 4 shows the dense depth maps obtained by different methods. The second and third rows represent the generated Pseudo-LiDAR point clouds from two perspectives. In the last row we zoomed in the local area for better display. The complete network generates the most realistic details and distributions of the intermediate point cloud.

### 4.3. Comparison Results

Because PLIN is the first work for point cloud interpolation, we only compare our method with the traditional interpolation method that averages the two consecutive depth maps and the state-of-the-art video interpolation network Super Slomo [18]. Note that, we retrain the Super Slomo network using the KITTI depth completion dataset [25].

#### 4.3.1. Quantitative Comparison

Table 2 reports the quantitative evaluation results of different methods. The traditional method averages two consecutive depth maps to obtain an intermediate depth map. However, the pixel values are relatively sparse and there is no obvious correspondence, so that the traditional method is not suitable for the interpolation of point clouds. Moreover, the video interpolation network [18] cannot handle the point cloud interpolation problem due to the challenging motion perception on the sparse depth map. Compared with these two methods, the proposed PLIN network is specially designed for the point cloud interpolation task and jointly guided by the explicit motion and realistic scenes, achieving the best performance.

#### 4.3.2. Visual Comparison

For visual comparison, we show three interpolated results achieved by different methods in Figure 5. In the first three rows of Figure 5, the color image as the current scene is depicted in the first column. In column 2 ground truth point clouds are presented, and the last three columns represent Pseudo-LiDAR point clouds obtained by super slomo, traditional method, and our method, respectively. In the last line we show enlarged local area for better viewing. As illustrated in Figure 5, suffering from the plain average interpolation, the traditional method generates fake objects which do not exist in the original scene. For Super Slomo [18], it is difficult or infeasible to learn the optical flow between depth maps. Therefore, Super Slomo produces disordered point clouds due to its the insufficient learning capability on motion and scenes. Compared with these methods, our method learns optical flow information of color images. Since our method takes into account scenes and appropriate motion information to guide the generation of Pseudo-LiDAR point clouds, the whole distribution of Pseudo-LiDAR point cloud is more similar to that of the ground truth point cloud.

## 5. Conclusions

In this paper, we have proposed a network to generate both temporally and spatially high-quality point cloud sequences. The main goal of the proposed method is to increase the frequency of LiDAR sensor data to alleviate the low frequency limitations of LiDAR sensors. In order to gradually perceive different modal conditions, we adopted a coarse-to-fine cascade structure. Specifically, the bidirectional optical flow explicitly guides consecutive sparse depth maps to generate an intermediate depth map, which is further improved by the warping layer. To obtain more accurate and dense depth information, the scene guidance module exploits the intermediate color image to refine the coarse depth map. Experimental results demonstrate that PLIN achieves promising performance on the KITTI dataset. The proposed model reasonably recovers the original intermediate 3D scene and outperforms other interpolation methods. To the best of our knowledge, this is the first deep framework for Pseudo-LiDAR interpolation, which increases the frequency of LiADR sensor data and shows appealing applications for more efficient multi-sensor systems. For autonomous driving systems, high-speed application scenarios will become a trend. Our method provides a method and way to improve the frequency of LiDAR sensor data, and further pursuit the high-quality synchronized perception of a multi-sensor system. However, the accuracy of our method still needs to be improved. Future work will involve efforts to build network structures, design loss functions, and better feature representations.

## Figures and Tables

**Figure 1 sensors-20-01573-f001:**
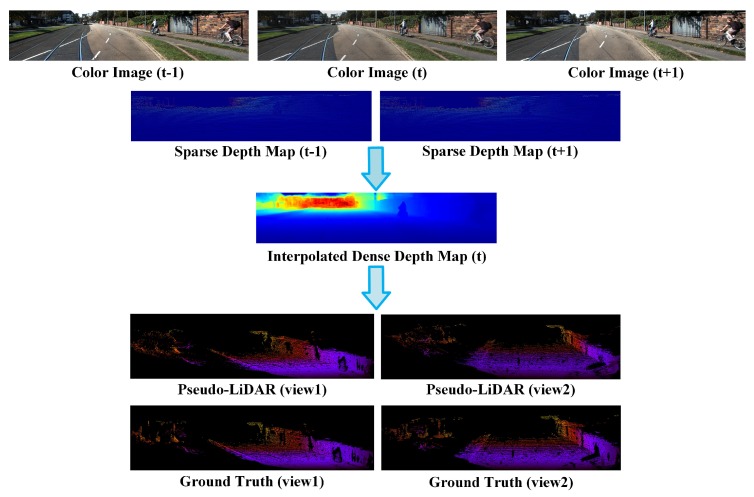
Overall pipeline of the proposed method. PLIN aims to address the mismatching problem of frequency between camera and LiDAR sensors, generating both temporally and spatially high-quality point cloud sequences. Our method takes three consecutive color images and two sparse depth maps as inputs, and interpolates an intermediate dense depth map, which is further transformed into a Pseudo-LiDAR point cloud using camera intrinsics.

**Figure 2 sensors-20-01573-f002:**
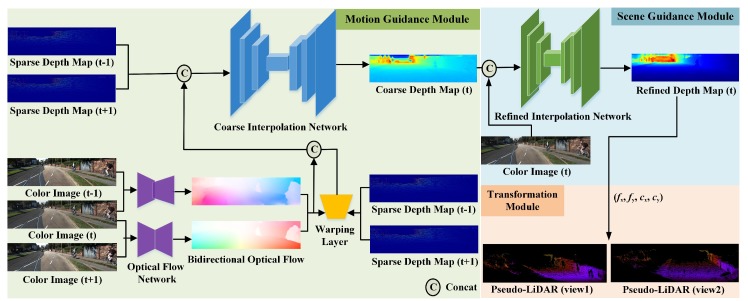
Overview of the proposed Pseudo-LiDAR interpolation network (PLIN). The whole architecture consists of three modules, including the motion guidance module, scene guidance module and transformation module.

**Figure 3 sensors-20-01573-f003:**
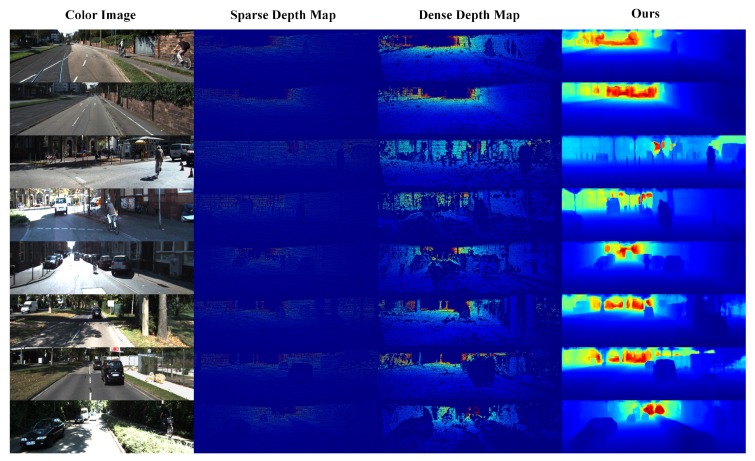
Results of interpolated depth map obtained by PLIN. From left to right, the color image as input is depicted in the first column. In column 2 the sparse depth map corresponding to the color image is presented, in column 3 the dense depth map represents the ground truth of network training. Finally, the result of our network prediction is depicted in column 4. Our method can recover the original depth information and generate much denser distributions.

**Figure 4 sensors-20-01573-f004:**
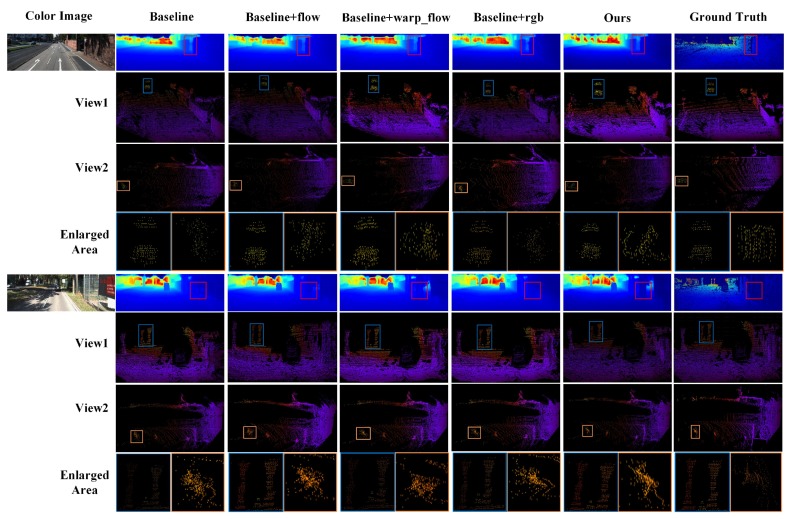
Visual results of the ablation study. We show the color image, interpolated dense depth map, two views of the generated Pseudo-LiDAR, and enlarged areas. The complete network produces more accurate depth map, and the distribution and shape of Pseudo-LiDAR are more similar to those of the ground truth point cloud.

**Figure 5 sensors-20-01573-f005:**
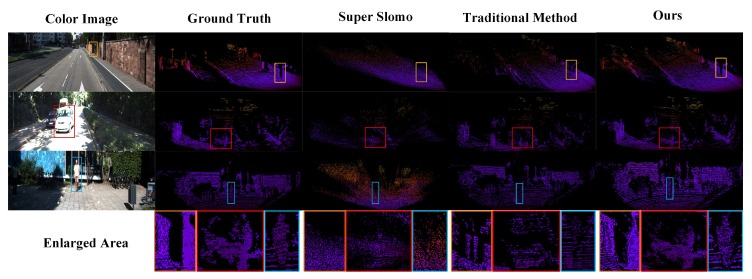
Visual comparisons of the point cloud obtained by different methods. We show the intermediate color images, ground truth, and interpolation result of Pseudo-LiDAR point clouds by three methods. Our model produces outlines and boundary regions for small objects (such as cars and people) that are more similar to ground truth.

**Table 1 sensors-20-01573-t001:** Ablation study: performance achieved by our network with and without each module.

Configuration	RMSE	MAE	iRMSE	iMAE
Baseline	1408.80	513.06	7.63	3.01
Baseline+flow	1335.06	514.4	8.04	3.47
Baseline+warp_flow	1216.63	532.84	9.03	4.04
Baseline+rgb	1238.25	495.24	6.38	2.95
Ours	1168.27	546.37	6.84	3.68

**Table 2 sensors-20-01573-t002:** Quantative evaluation results of the traditional interpolation method, Super Slomo [18], and our method.

Method	RMSE	MAE	iRMSE	iMAE
Traditional Interpolation	12,552.46	3868.80	-	-
Super Slomo [18]	16,055.19	11,692.72	-	-
Ours	1168.27	546.37	6.84	3.68

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
