# Peer review of "PLIN: A Network for Pseudo-LiDAR Point Cloud Interpolation"

_sensors, 2020, doi:10.3390/s20061573_

Round 1
Reviewer 1 Report
- The manuscript is prepared with high professions. Either the editorial and technical contents are all well-organized. Although the manuscript can be accepted as it is, I still would have a few comments, however, to better help readers’ comprehension of this work.
- Page 2, section 1: I would suggest the authors to address more on why “directly working on the 3D space and generating a new point cloud is challenging”.
- Some section titles are all capitals but others are not – please fix them.
- Page 5-7: it is better to illustrate, elucidate, or clearly introduce the structure of Hb(eq. 1), Hf(eq. 2), Hm (eq. 7), and Hs (eq. 10) as they are the key parts of the proposed PLIN.
- Page 9, Table 1: how those numbers were obtained? For example, how the RMSE is defined in this case.
- Page 7, line166-167: It is still not clear how the weight w1 and w2 were selected. I suggest the authors to provide/explain the selection basis so as to favor interested readers.
- Page 9, Figure 4: please fix the sequences of the shown results (top to bottom) as: color image, view 1, view 2, enlarged area
- Readers would expect to see more than one example of the ablation study in section 4.2, and certainly the consistency and stability in difference cases.
- Page 10, results of Figure 5: to my (and many readers’) best vision, I don’t see the significant evidences to conclude the caption “our model produce more sharp outlines and boundaries for small objects such as cars and people”. Some other means of evaluation should be supplemented.
- Page 11: This study shows a great deal of work (and efforts) on Pseudo-Lidar interpolation. I believe a more detailed conclusions/ significances of the study/ impact to field applications/ further studies would be very much appreciated by interested readers.
Reviewer 2 Report
This paper presents a LiDAR interpolation method using deep CNN. The method is well presented and experiments show that the performance is well enough for practical use. However, it is vague that the final form of the result is either in 3D point cloud or in 2D dense depth map. And there are no explanations of the included figures in the manuscript. Authors should include explanations of every figure in the manuscript along with the figure captions. For example, Figure 3 is confusing: what are the source of spare depth map and dense depth map? Are they ground truth? Overall, I think the method is novel and practical, well presented, and of interest to the readers.
Round 2
Reviewer 1 Report
Hello authors, I appreciate your efforts that have made to the revision. Please correct the repeating use of “on the other hand” in Page 2 Line 31-33. I have no further question/comment after this.